# Lipid Changes in the Peri-Implantation Period with Mass Spectrometry Imaging: A Systematic Review

**DOI:** 10.3390/life13010169

**Published:** 2023-01-06

**Authors:** Stefánia Gitta, László Márk, József L. Szentpéteri, Éva Szabó

**Affiliations:** 1Department of Analytical Biochemistry, Institute of Biochemistry and Medical Chemistry, Medical School, University of Pécs, 7624 Pécs, Hungary; 2National Human Reproduction Laboratory, University of Pécs, 7624 Pécs, Hungary; 3MTA-PTE Human Reproduction Research Group, University of Pécs, 7624 Pécs, Hungary; 4Institute of Transdisciplinary Discoveries, Medical School, University of Pécs, 7624 Pécs, Hungary

**Keywords:** animal model, embryo, implantation, lipids, mass spectrometry imaging, systematic review

## Abstract

Mass spectrometry imaging is a sensitive method for detecting molecules in tissues in their native form. Lipids mainly act as energy stores and membrane constituents, but they also play a role in lipid signaling. Previous studies have suggested an important role of lipids in implantation; therefore, our aim was to investigate the lipid changes during this period based on the available literature. The systematic literature search was performed on Ovid MEDLINE, Cochrane Library, Embase, and LILACS. We included studies about lipid changes in the early embryonal stage of healthy mammalian development published as mass spectrometry imaging. The search retrieved 917 articles without duplicates, and five articles were included in the narrative synthesis of the results. Two articles found a different spatial distribution of lipids in the early bovine embryo and receptive uterus. Three articles investigated lipids in mice in the peri-implantation period and found a different spatial distribution of several glycerophospholipids in both embryonic and maternal tissues. Although only five studies from three different research groups were included in this systematic review, it is clear that the spatial distribution of lipids is diverse in different tissues and their distribution varies from day to day. This may be a key factor in successful implantation, but further studies are needed to elucidate the exact mechanism.

## 1. Introduction

Mass spectrometry imaging (MSI), a rapidly growing technique in mass spectrometry (MS), allows molecular signals to be directly detected from the surface of thin biological tissues [1,2]. This method directly provides information about the molecular characteristics of the sample, based on the ‘mass-to-charge (m/z) ratio’ of the molecules. In addition, it can also be paired with tissue localization, which the MSI software can convert into a 2D or even 3D image [3]. Moreover, when combined with other imaging techniques, it can be even more powerful, moving towards multimodal imaging [4].

Imaging is a technique that does not require a molecule-specific labeled sample or prior molecular knowledge of the tissue being analyzed [5]. It is unique in that it can cause the peptides, proteins, oligonucleotides, lipids, sugars, and other small molecules in biological samples to become visible, making it a useful tool for molecular mapping [3]. In contrast to conventional MS, where the spatial information of the given molecule is lost after sample preparation, homogenization, and extraction methods, with MSI, no extraction and separation is needed, and the spatial distribution of these molecules can be studied. However, using MSI alone does not provide us with information about the molecule and its structure, since only m/z ratios and their distribution on the investigated sample or tissue are obtained. This is a major limitation of this technique because it is not suitable for the structural identification of biomolecules from a complex biological or clinical tissue section without fragmentation. On the other hand, its combination with different separation techniques, on-tissue enzymatic digestion, and/or using tandem MS can overcome this problem. Precursor ion selection, isolation, and fragmentation directly from the tissue sections is possible by using MALDI coupled with TOF/TOF, Q-TOF, Orbitrap, and tandem mass spectrometers. Moreover, enzymatic digestions of the proteins on the tissue section surface can be carried out by using bottom-up proteomic workflow combined with MSI. Therefore, it can be suitable for the discovery of biomarkers in certain circumstances, since MSI combined with other techniques for the fragmentation and identification of molecules could be used to predict the relative concentration of a given molecule in tissue or even in a group of cells. However, it should be mentioned that intensities on different MSI images cannot be directly compared due to several causes, e.g., differences in ionization or matrix effects [4]. In addition to the accumulation of a biomarker in a particular tissue fragment, temporal characteristics can be obtained from the molecule. By looking at samples from several different time points, a period of time during which a particular molecule is expressed (e.g., during embryonic development or tumor development) can be observed [3,5]. Lipids have a wide range of functions in the human body. In the form of triacylglycerols (TAGs) they serve primarily as a reserve nutrient, while amphipathic phospholipids (PLs) are involved in the construction of cell membranes. Glycerophospholipids are the main building blocks of cell membranes with a distinctive asymmetry on the two (intra- and intercellular) sides of the cell membrane, as well as between the different cell organelles [6,7]. Sphingosines are another large group of lipids including ceramides, gangliosides, and sphingomyelin that play an important role, among others, in apoptosis, cell growth, differentiation, and adhesion [8]. As membrane constituents, these lipids participate in endo- and exocytosis [6,8] and regulate chemotaxis and cytokinesis [9]. They also regulate the cellular processes, directly or indirectly, that are involved in cell growth, synaptic signaling, or even the monitoring of the immune system [6,8,9,10]. Fatty acids (FAs) very often underlie the biological functions of lipids: TGs contain three, whereas phospholipids have one or two fatty acids. These FAs usually have an even number of carbon atoms and can be divided into saturated and unsaturated FAs according to the number of double bonds in the chain. Saturated FAs increase the rigidity of the cell membrane, while unsaturated fatty acids increase fluidity. Some FAs, mainly those with 20 carbon atoms, are involved in certain signaling processes. Eicosanoids are oxygenated derivatives of these FAs involved in platelet aggregation, the induction or resolution of inflammation, the regulation of immune responses, and smooth cell contraction, among others [7,10].

There are many articles on the MSI analysis of peptides and proteins [3,5,11,12,13,14,15]; however, due to the more difficult identification of lipids, especially PLs, we are lacking such articles. Last year, almost 600 articles on protein MSI were published in the PubMed database, while the number of lipidomic MSI articles was less than 100. PLs are a large group of lipids with a wide variety of components resulting in different ionization profiles (e.g., phosphatidylcholines (PC)), typically ionized in the positive, while phosphatidylserines (PS) in negative mode and several lipids with the same m/z ratio can belong to a large number of diverse PLs [2].

The most widely used MSI techniques for lipid determination in biological tissues are desorption electrospray ionization (DESI), matrix-assisted laser desorption ionization (MALDI) and secondary ion mass spectrometry (SIMS). A great advantage of these techniques is that the spatial distribution of individual lipids can be studied in 2D or even in 3D images [2,16]. Their application includes the lipid analysis in histopathological regions (e.g., atherosclerotic plaques [17], stenotic aortic valve [18], brain sections after focal cerebral ischemia [19]), the identification of potential biomarkers in tumors, as well as their potential use in diagnosis [20]. However, the MSI technique has several limitations, including the detection of phospholipid regioisomers. The fatty acids that comprise phospholipids often differ only in the localization of the single unsaturated bond (e.g., C18:1n-7 or C18:1n-9) or in the conformation of the unsaturated bond (cis and *trans* isomers of C18:1n-9: cis oleic acid and *trans* elaidic acid). A further problem is that the MSI technique does not provide information on the relative position (sn-1 or sn-2) of the fatty acids to the glycerol backbone, although the position of the fatty acid is biochemically very important [21,22]. Another limitation is the relatively low spatial resolution of these conventional MSI methods with pixel sizes between 30 and 100 μm [23]. Some recent techniques have attempted to overcome these limitations, for example, MALDI-2 with post-ionization increased a spatial resolution as small as 6 μm [23], resulting in a high spatial resolution of phospholipids with a 2′,5′-dihydroxybenzoic acid (DHB) matrix [24]. Another relatively new technique is the ozone-induced dissociation (OzID) with MALDI-MSI, where the monounsaturated isomers differ only in the position of the double bond (namely n-7 or n-9 group), and the localization of double bonds in the sn-1 position gives additional information to phospholipid distribution in biological samples [25].

Infertility is a worldwide problem affecting over 186 million people, about 8–12% of couples in their reproductive age [26]. The effectiveness of assisted reproduction technologies (ARTs) around the world in 2012 led to approximately a 27% pregnancy rate per aspiration and almost 20% delivery rate per aspiration [27]. Although ART techniques are improving, the live birth rate per aspiration is still low, and there is wide variation among various regions of the world [28]. In the background, several factors could stand, such as impaired endometrial receptivity, asynchrony between the endometrial stage, and embryo development as a result of disturbed hormonal answers during ovarian stimulation. The role of lipid metabolism [29] and signaling [30] is also crucial; however, the exact mechanism is still not fully understood. Anandamide (N-arachidonoylethanolamide, AEA), a lipid-derived endocannabinoid hydrolyzed from the cell membrane component N-acylphosphatidylethanolamine, which binds to the cannabinoid receptors, can play a role in uterus receptivity and embryo implantation. An animal study proved that a low AEA concentration (14 nM) can promote blastocyst attachment, while a higher concentration causes a delay, suggesting the central role of AEA in embryo implantation [31]. AEA also plays a role in the receptive uterus: the highest levels can be measured in the nonreceptive uterus, while the lowest levels at implantation sites suggest protection from the detrimental effect of this mediator on the implanting embryos [32]. A human study in IVF patients also corroborated this, as high blood levels of AEA together with lower degradation enzyme activity was associated with implantation failure and early miscarriage [33]. Arachidonic acid-derived prostaglandins have also been shown to play a regulatory role in embryo implantation and uterine receptivity via angiogenesis stimulation and myometrium relaxation and contraction [29]. PGE_2_ and PGF_2α_ can be involved in maternal–embryonal crosstalk and successful implantation in both animal [34,35,36,37] and human studies [38,39].

Early embryonic development involves the period from fertilization to implantation. Spatial and temporal monitoring of this development allows a thorough examination of the series of special events required for differentiation. Since human embryos can only be tested to a limited maturity due to legal regulations, different animal models, such as mouse models, are used for this purpose. However, it should be noted that there are also significant differences between the two species in terms of their genes and their expression, leading to different biochemical pathways being activated during their embryonic development. Whatever animal model is used, the study of early embryonic development has many benefits. We can not only understand the biochemistry of a healthy germ cell and fertility, but we can also explain different pathologies. By modeling implantation, we can understand the cause of high rates of early pregnancy loss and improve the method of conception. This can be exploited mainly in the field of ART.

Since different lipids play a central role in membrane assembly and cell communication, they may also play an important role in implantation and early embryonal development. MSI, coupled with other techniques for identification, can show the spatial and temporal distribution and precise location of different lipids in tissues without labelling and lengthy prior preparation or tissue disruption. This powerful MS technique, combined with the precursor ion selection, isolation, and fragmentation process, provides the possibility of the structure identification of different kinds of molecular species. Therefore, our aim was to systematically review the available literature on lipid changes measured during the peri-implantation period using the MSI technique.

## 2. Materials and Methods

This systematic review was registered prospectively in PROSPERO, under: CRD42021279133. The methodology and the results are reported according to the PRISMA (Preferred Reporting Items for Systematic Reviews and Meta-Analyses) guidelines for systematic reviews (Appendix A) [40,41], as well as the guidelines of the Cochrane Handbook of Systematic Reviews and Intervention [42].

### 2.1. Eligibility Criteria

We included studies about mammalians, studies about animals other than mammalians (e.g., fish, birds), plants, and fungi, while in vitro cell studies were excluded. We included studies about early embryonal development after either normal or in vitro fertilization but excluded studies about oocyte or spermatozoa only. Studies about healthy embryonal development were included, but studies about unhealthy animals (embryo or mother or father), and animal models of somatic, metabolic, or genetic disease were excluded. No restriction on experimental study types was applied: we only excluded reviews, editorials, and comments not publishing the original data of authors. We included studies about lipid changes in the early embryonal stage publishing their data as mass spectrometry imaging. Only complex lipids were included, while small, lipid-derived molecules (e.g., prostaglandins, thromboxanes, leukotrienes), receptors, and proteins involved in lipid metabolism or signaling were excluded.

### 2.2. Literature Search

The electronic literature search was performed on the following databases from the inception of each database up to September 2021: Embase, Cochrane Central Register of Controlled Trials (CENTRAL) Ovid MEDLINE, and Latin America and the Caribbean Literature on Health Science (LILACS). No language restriction was used. The search strategy was developed with terms related to zygote, fertilization, implantation, lipids, lipidomics, and mass spectrometry imaging. The search strategy on Ovid MEDLINE was as follows: (zygot*.mp OR blastocyst.mp OR fertilization.mp OR embryo.mp OR preimplantation.mp OR periimplantation.mp OR “implantation site”.mp OR mesometrial.mp OR antemesometrial.mp OR ante-mesometrial.mp) AND (lipid.mp OR lipids.mp OR lipidomic.mp) AND (imaging.mp OR MALDI.mp OR MSI.mp). The detailed search strategy with results for Ovid MEDLINE is available in Appendix A.

We manually searched the references of included articles and related reviews for potentially relevant articles. We also searched grey literature for congress abstracts that might be relevant, as well as Latin America and the Caribbean Literature on Health Science (LILACS). All citations were then combined, and duplicates were excluded.

### 2.3. Study Selection, Risk of Bias Assessment

Two review authors (E.S., G.S.) independently screened the abstract, title, or both of every record to determine potentially relevant articles. The abstract screening was performed using an online program (http://abstrackr.cebm.brown.edu, 25 October 2021) [43]. Then, the two reviewers independently screened the full-text articles for inclusion/exclusion criteria on Rayyan.ai [44]. Disagreements between the reviewers were resolved by further discussion until consensus. The Risk of Bias (RoB) was also assessed independently by the two authors (E.S., G.S.), according to the SYRCLE RoB tool for animal studies [45].

### 2.4. Data Extraction and Synthesis

Two authors (E.S, G.S.) independently extracted data from the included articles. From full-text publications, we extracted data on the first author, year of publication, investigated animals, type of fertilization, embryonal age, used technique(s), and determined complex lipids. 

## 3. Results

The search was run on 15 September 2021 and retrieved 1093 unique records. After removing duplicates, 917 records were screened based on their title and abstract. Most articles were excluded because they did not meet the inclusion criteria (Figure 1), so 20 full-text articles were further evaluated, and finally, five studies met our inclusion criteria and were included in this systematic review.

Of the 20 potentially relevant articles, one was an in vitro experimental study [46]; three articles were excluded because of the wrong population (cumulus cells [47] or older fetus [48,49]); and 10 published either no MSI results, only MS data [50,51,52,53,54,55,56,57,58], or only proteomics of the embryos [59]. One abstract was also excluded because no data extraction could be performed [60].

The main characteristics of the included studies are shown in Table 1. Three studies investigated lipid changes in mice [61,62,63], while two other studies investigated bovine embryos [64,65]. Of the three mouse studies, two used CD-1 mice [61,62], and in the third study, an experimental model of spontaneous preterm birth (deletion of uterine Trp53 [p53^d/d^]) was used with normal controls (p53^f/f^) [63]. Only one study published the MSI data of fertilized embryos (two-cell and eight-cell) [64] and one study the receptive uterus after induced ovulation [65], while mice studies investigated the whole uterus with implanting embryos [61,62,63]. Burnum et al. [61] published a day-to-day approach in the early implantation period (4–8 days), and Lanekoff et al. on day 6 (implantation) [62] and on day 8 (decidualization) [63]. Regarding the various MSI techniques, three studies used MALDI [61,64,65], while Lanekoff et al. [62,63] used nano-DESI for lipid imaging in the peri-implantation period.

Most of the studies had an unclear risk of bias (Appendix A), while two studies had a high risk of performance bias (because of no blinding).

### 3.1. Lipids in Embryos Early after Fertilization

In the two- and eight-cell stages (Table 2) there was a clear difference between zona pellucida and blastomeres [64]. In the zona pellucida, the two lipids in the highest concentration were phosphatidic acid (PA) PA(20:0/20:3) and phosphatidylethanolamine (PE) PE(24:0/20:0) in both time points. In contrast, the two main lipids in the blastomeres were PE(16:0/18:1) and Ceramide (Cer) (Cer(18:1/22:0).

### 3.2. Lipids in the Receptive Uterus

In a mouse model, on day 4 (Table 2) before embryo attachment [61] in the luminal epithelial cells, a relative increase in sphingomyelin (SM16:0) and phosphatidylcholines (PCs) with mainly unsaturated fatty acid chains (PC34:0 (16:0/18:0), PC34:1 (16:0/18:1), PC36:1 (18:0/18:1), PC34:2 (16:0/18:2), PC36:2 (18:0/18:2), PC36:4 (16:0/20:4), PC38:4 (18:0/20:4)) was seen, while PC32:0 (16:0/16:0) and PC40:6 (18:0/22:6) were almost undetectable. In the negative mode, only two phosphatidylinositols (PI), PI34:0 (16:0/18:0) and PI40:6 (18:0/22:6) were increased, while PE34:1 (16:0/18:1) decreased compared to other cells.

In cattle [65], the luminal epithel of the receptive uterus showed a clear accumulation of PC35:2, while PC38:7, PC38:5, and PC38:4 showed great accumulation in the uterus region compared to the myometrium. In contrast, SM34:1 was mainly accumulated in the myometrium. Other lipids (SM34:2, PC31:0, PC32:0, PC47:0, PC40:6) were diffusely distributed in the whole uterine section.

### 3.3. Lipids in Implanting Embryos and Uterus

On day 5 (Table 2) at the site of embryo attachment [61] in uterine stromal cells, most glycerophospholipids were increased, except for PC32:0 (16:0/16:0), which was almost undetectable. Published PCs were only increased at the site of embryo attachment, but in other cells they were not; glycerophospholipids in the negative mode were not selectively increased in these cells, except for PI34:2 (16:0/18:2), PI36:2 (18:0/18:2), and PI40:6 (18:0/22:6), but these were only increased around embryo attachment and not in other uterine cells.

On day 6 [61] in the primary decidual zone (PDZ), the glycerophospholipid expression was the most intense, except for PC32:0 (16:0/16:0). The investigated PC lipids showed the most intense expression in the PDZ (SM16:0 (-/16:0)) or around this zone (almost all PC lipids), while PC34:1 (16:0/18:1) was the most intense in the mesometrial (M) pole. The expression of PE34:1 (16:0/18:1) and PI38:5 (18:1/20:4) was also increased in the M pole, while PI34:2 (16:0/18:2) and PC36:2 (18:0/18:2) were the most intense in the antemesometrial (AM) pole. Other glycerophospholipids showed a less clear accumulation but were also found around the PDZ. Another study [62] also found an increased PC36:2 in the AM pole. The fatty acid 18:2 was increased in the PDZ, while the lysophosphatidylcholine (LPC) LPC18:0 was decreased.

Similarly, on day 7, PC32:0 (16:0/16:0) was intense in the uterus, except for the PDZ [61]. PC34:0 (16:0/18:0), PC34:1 (16:0/18:1), PE34:1 (16:0/18:1), and PI38:5 (18:1/20:4) showed an increased intensity in the M pole, while glycerophospholipids containing one fatty acid with at least two double bonds (PC34:2 (16:0/18:2), PC36:2 (18:0/18:2), PC36:4 (16:0/20:4), PC38:4 (18:0/20:4), PC40:6 (18:0/22:6), PI34:2 (16:0/18:2), PI36:2 (18:0/18:2), PS36:2 (18:0/18:2), PI40:6 (18:0/22:6)) showed a higher expression, mainly in the AM pole.

On day 8 [61], most of the phospholipids could be divided into two groups according to their location. In the M pole, a high expression was seen in those PC, PE, phosphatidylglycerol (PG), and phosphatidylserine (PS) lipids that contained one C18:1, while in the AM pole these lipids showed a very decreased accumulation. In contrast, PC, PI, and PS lipids containing C18:2, C20:4, or C22:6 as polyunsaturated fatty acids showed an increased accumulation in the AM pole but almost no intensity in the M pole.

In another study [63], very similar results were found in mouse implantation sites on day 8 of pregnancy: the M pole was the primary localization site for PCs containing fatty acids with one double bond (PC 34:1, PC36:1 (18:0/18:1), PC 36:3 (18:1/18:2)), while PCs containing more double bonds were localized on the opposite site, the AM pole (PC36:2 (18:0/18:2), PC38:4, PC40:6). Several diacylglycerols (DGs) were also found with higher intensity on the AM pole (DG34:2, DG36:4, DG36:3, DG36:2, DG38:4).

## 4. Discussion

In this study, we reviewed the available articles investigating lipids in the peri-implantation period using MSI techniques. Although the topic is highly current, only five publications were found in the systematic literature search that met all our inclusion criteria. Several former studies investigated the role of lipids in oocytes [66,67], early stages of embryo development [52,56,58,66,68,69,70,71], placenta in late pregnancy [72,73], as well as endometrial tissues during the window of implantation [74], and in endometrial fluid [75]. However, in these studies, lipid composition was mostly investigated with MS and not with MSI. The main advantage of this method compared to conventional MS methods is that lipids can be analyzed in space without extensive preparation and without destroying the section.

In recent years, MSI has widely been used for lipidomics for detecting possible biomarkers in brain, liver, oral, breast, colorectal, skin, or other cancers [2,20,76]; in brain and neuronal tissue research [77] and models of fetal or adult cerebral ischemia [19,78,79,80]; in neurodegenerative disorders [81] such as Alzheimer’s disease [82] or Niemann-Pick disease [83]; as well as in cardiovascular diseases [84] such as atherosclerosis.

Although MSI could be a very useful tool for monitoring lipid changes around implantation and in the whole perinatal period, only a very small number of studies have used the MSI technique to monitor embryonic or fetal development. A review on the usefulness of the MSI technique and its potential importance in fertility research was published as early as 2012 [85], but the number of publications on fertilization, implantation, and early embryonic development in animal models of human pregnancy has not increased dramatically over the last decade. Most publications report MSI sections of ovaries or oocytes from different animals and investigate or compare the different spatial occurrences of different lipids or discuss similarities/differences between species. The ovaries are the site of oocyte development and maturation. Therefore, the study of lipidomic abnormalities in the ovaries bears particular importance for fertility. MSI sections of banana shrimp (*Penaeus merguiensis*) [86], mosquito (*Aedes aegypti*) [87], porcine [88], and bovine [67] ovaries showed different accumulations of lipids in the different compartments, suggesting metabolic differences between cell types which may be due to their different functions in oocyte maturation (energy production and/or storage, membrane synthesis or signaling). The lipid distribution of oocytes was also investigated by MSI in African clawed frog (*Xenopus laevis*) [89], crustaceans (*Gammarus fossarum*) [90], mice [91], and cattle [59].

Zebrafish is a widely used model for embryonic development, and the lipid changes during early embryonic development have already been investigated with the MSI technique [92,93]. Another possible model for early embryonic development is the *Xenopus laevis*, in which the spatial and temporal variation of lipids has also been studied using time-of-flight secondary ion mass spectrometry (ToF-SIMS) imaging [94]. There are also some studies investigating fetuses at the end of pregnancy, such as the study about a 50-day-old whole pig fetus [48,95]; a 14-day-old [96] and a 17.5-day-old whole mouse fetus [49]; or only lung samples of a mouse fetus at day 19 of gestation [97].

In a human study, Yamazaki et al. [98] showed that even placental pathologies due to maternal (gestational hypertension or preeclampsia) or fetal (small for gestational age) diseases may disturb the lipid composition in the terminal and stem villi at delivery. In a mouse study, a US research group [99] showed that maternal diet and obesity strongly influence the lipid composition of the placenta of mice on gestational day 12.5: there was a 2.5 to 6-fold increase in PC36:1, PC38:3, LPC16:1, LPC18:1, and LPC20:1 in the decidua of the obese group following a Western diet compared to lean mice on a normal diet.

Although there is growing evidence to support the role of lipids and lipid-derived mediators in the peri-implantation time, the underlying mechanisms are not fully understood. Some important lipid metabolites in early pregnancy are the endocannabinoid anandamide (AEA), arachidonic acid-derived prostaglandins, lysophosphatidic acid, and sphingolipids [29]. Therefore, there should be a link between the lipid composition of the cells and lipid signaling during early embryonal development and implantation [30].

Previous studies have shown that the major lipids of different mammalian oocytes are quite similar, with PC34:1 being the major component, but using principal component analysis, the lipids of each species are very distinct [66,100]. Therefore, different animal models may be suitable for tracking large changes in the human peri-implantation period, but subtle changes may differ substantially between species. Thus, the human use of potential biomarkers seen in animal models is possible only with some criticism.

In our systematic review, we found only one study investigating the lipid composition of pre-implantation embryos in the two- and eight-cell stages with MSI. In both time points, there was a clear difference between lipid localization in the embryo: blastomeres showed a higher abundance of PE(16:0/18:1) and Cer(18:1/22:0), while the zona pellucida showed a higher amount of PA(20:0/20:3) and PE(24:0/20:0). Former studies have already shown that the lipid composition of the developing embryo changes, so the concentration of lipids relative to each other and the most abundant lipids vary almost from day to day [52,59,101]. In a mouse study, the unfertilized oocyte had a more simple lipid composition with PC34:1 being the most abundant lipid on DESI-MS in negative mode; in the two- and four-cell embryos, more lipids were mainly determined with 34–38 carbon atoms, and one, two, or five double bonds, while the blastocyst had a higher relative abundance of PC lipids, with PC36:2 as the most abundant lipid [101]. Similar differences were found in TG lipids in bovine oocytes (most abundant: TG52:3) and blastocysts (most abundant: TG52:1) containing different lipids with varying relative abundances in the DESI-MS positive mode [102]. Not only do the individual lipids change during early embryonal development, but the relative abundance of the different lipid classes change as well. Blastocysts have the highest intensity of most lipids (PC, PE, PG, and PI), compared to two- and four-cell embryos: only short- and medium-chain PS lipids were found in higher abundance in the two- and four-cell embryos compared to blastocysts [103].

There is not only a dynamic change in the relative abundance of different lipid species during embryo development, but also in their relative fatty acid content as well [69]. Blastocysts had higher oleic acid (C18:1n-9, OA) and linoleic acid (C18:2n-6, LA) levels compared to the ≤6 cell stage, and in the first days of embryo development (two-cell to blastocyst) with ^13^C-labelled fatty acids, a selective accumulation of LA (but not palmitic acid (C16:0)) was detected [69]. In this review, we also found that the AM-pole (site of implantation) had a higher intensity of LA-containing lipids than the M-pole. This supports the possible role of unsaturated fatty acids in early embryonic development and implantation.

There are some differences among the main lipids in in vivo- and in vitro-fertilized blastocysts [51,52,102]. Not only bovine species (Simmental or Nellore), but also the origin (in vivo or in vitro) of embryos affected the lipid composition: in vivo embryos had a significantly higher abundance of PC32:1, PC34:2, and PC36:5, while in vitro embryos of both species showed a higher abundance of PC32:0 [56]. In contrast, Annes et al. [51] found a significantly lower abundance of PC32:1, PC32:0, and PC34:2 in in vivo-produced bovine embryos compared to two groups of in vitro-derived blastocysts, while the relative abundance of PC36:2 was higher in in vivo blastocysts. Phospholipids and fatty acids also differ: in in vivo-derived blastocysts the most abundant one was oleic acid (C18:1n-9), while in in vitro-produced blastocysts this was palmitic acid (C16:0) and stearic acid (C18:0) [102]. In addition, IVF can not only affect the embryo before implantation, but can also cause longer-lasting changes in the lipid composition of maternal tissues such as the placenta. In a previous study [73], IVF placentas were heavier and contained less neutral lipids (cholesterol esters, TG) but higher concentrations of phospholipids (PC, PE, PI, PG). These phospholipids are the major sources of fatty acids, which are important precursors of signaling molecules, so it is interesting that there were significant differences between the relative composition of placentas from the two groups. Most investigated PC (e.g., PC32:0, PC34:2, PC36:4, PC36:3, PC36:2, PC36:1, PC38:4) and PE species (e.g., PE36:2, PE38:5, PE38:4, PE40:6 PE40:4), as well as several cardiolipins, PGs, PIs, and PSs, were increased in the IVF group, while only some of them were significantly decreased. This may be due to the reduced expression of proteins regulating phospholipid biosynthesis [73].

Apart from the type of insemination (in vivo or in vitro), the conditions used during IVF can also affect the lipid composition of the developing embryo. The low O_2_ concentration of the incubator in bovine serum albumin (BSA) culture medium altered the lipid composition of the embryo, and PC36:1 became one of the major components, whereas, at higher O_2_ concentrations, PC34:1 was the other major lipid component besides PC 36:1. The use of fetal calf serum (FCS) culture medium instead of BSA caused the accumulation of SM16:0, so the fatty acids constituting the lipids (C18:1 or C16:0) were modified depending on the culture conditions; however, the culture media (BSA or FSC) had only a small effect at a low O_2_ concentration [66]. Although in this study the fatty acid composition and lipid content of the culture medium were not investigated, a mouse model suggested that the higher lipid content of the culture medium or the different fatty acid composition (higher saturated and unsaturated fatty acid content) may disturb the developing embryo, causing delayed blastocyst development, a lower implantation rate, and altered fetal outcome [104]. These adverse effects may be due to higher superoxide levels and different lipid accumulation within cells caused by different lipid content and fatty acid composition [104].

As discussed in detail above, it is clear that, on the one hand, the mode of fertilization and early developmental conditions can greatly influence the lipid composition of the embryo and endometrium during the peri-implantation period, and on the other hand, these variations in lipid composition can have longer-term effects on the developing embryo/fetus and placenta through a number of secondary messengers. Therefore, it is important to understand the spatial and temporal changes in lipids that normally occur during the peri-implantation period, in order to identify key components that predict the upcoming changes and their direction for the cells. Inadequate materno-fetal communication, in which lipids are one of the underlying factors, can lead to impaired implantation and early miscarriage, which may be partly responsible for the still low success rate of IVF.

The major strength of this study is the systematic literature search in four databases. It is also important that the whole process of screening and data extraction was carried out following the principles of evidence-based medicine. Although for ethical reasons, no human studies were conducted on this topic, the inclusion of studies in exclusively mammalian models makes our results applicable with good approximation to the changes around the human peri-implantation period.

The major limitation of the present study is the very small number of studies included in this systematic review. This is probably due to the more difficult investigation of the peri-implantation period. While the lipid composition of oocytes and zygotes has been reported in many different articles, the experimental background required to study the peri-implantation time is much more complex. Another limitation is that the studies on mouse embryos and receptive uteri are from three different publications by two research groups, and several authors are co-authors on all three papers. This is probably due to the highly specialized nature of the subject, and the fact that lipid analysis with MSI requires expensive equipment and specialized knowledge. On the other hand, the method used for mass spectrometry imaging differs among these three articles: while Burnum et al. [61] used MALDI-FTICR as imaging, in the two articles by Lanekoff et al. [62,63], nano-DESI MSI was used. Based on the findings of these five articles, only limited conclusions can be drawn about the possible role of lipids in the peri-implantation period, and further well-designed MSI studies are needed to find possible lipid biomarkers in this period.

As discussed above, lipid mediators (e.g., AEA, PGE_2_) have been shown to play a prominent role in the peri-implantation period. If we could identify both the spatial and temporal change in these or other lipid mediators and link them to specific tissues or cells in different tissues, we could be closer to understanding the biochemical mechanism of implantation from both maternal and embryonal sites. In addition, high-resolution MSI could also be used to discover new lipid-derived biomarkers that can predict either early implantation success or miscarriage, opening up new treatment options. This also might help to increase the low success rate in IVF techniques and give hope to couples struggling with infertility.

## 5. Conclusions

In conclusion, the peri-implantation period is characterized by a number of lipid changes, both embryonic and maternal, and further MSI studies are needed to map the exact temporal and spatial variation of these changes.

## Figures and Tables

**Figure 1 life-13-00169-f001:**
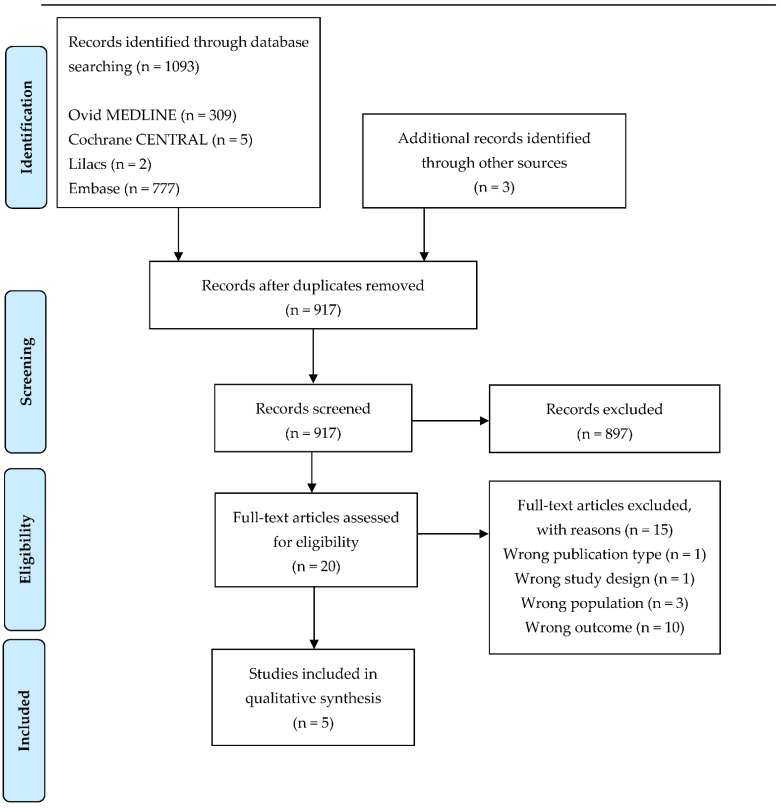
Flow diagram of the study.

**Table 1 life-13-00169-t001:** Characteristics of included studies.

Author, Year of Publication	Investigated Animals	Type of Fertilization	Age of Embryos	Type of Samples	Imaging Method
Burnum KE, 2009 [61]	CD-1 mouse	Normal	Day 4, 5, 6, 7, 8	Implanting embryos with uterus	MALDI-TOF, MALDI-FTICR
Lanekoff I, 2015 [62]	CD-1 mouse	Normal	Day 6	Implanting embryos with uterus	Nano-DESI MSI
Lanekoff I, 2016 [63]	p53^f/f^ vs. p53^d/d^ mouse	Normal	Day 8	Implanting embryos with uterus	Nano-DESI MSI
Ferreira MS, 2014 [64]	Cattle	IVF	Day 2–3	2-cell, 8-cell embryo	MALDI-LTQ-XL
Belaz KRA, 2016 [65]	Cattle	IVF	Day 4, 7	Uterus after induced ovulation	MALDI-MS

FTICR: Fourier-transform ion cyclotron resonance; IVF: in vitro fertilization; LTQ: linear trap quadrupole, MALDI: matrix-assisted laser desorption/ionization; MSI: mass spectrometry imaging; nano-DESI: nanospray desorption electrospray ionization; TOF: time-of-flight.

**Table 2 life-13-00169-t002:** Lipids potentially involved in the peri-implantation period based on mass spectrometry imaging data.

Time	Place	Lipid	Result
2-cell stage (bovine) [64]	Blastomere	PE(16:0/18:1); Cer(18:1/22:0)	↑
	Zona pellucida	PA(20:0/20:3); PE(24:0/20:0)	↑
8-cell stage (bovine) [64]	Blastomere	PE(16:0/18:1); Cer(18:1/22:0)	↑
	Zona pellucida	PA(20:0/20:3); PE(24:0/20:0)	↑
Receptive uterus (bovine), day 7 [65]	Myometrium	SM34:1	↑
	Luminal epithel	PC35:2	↑
	Uterus region	PC38:7; PC38:5; PC38:4	↑
	Whole section	SM34:2, PC31:0, PC32:0, PC47:0, PC40:6	↑
Receptive uterus (mouse), day 4 [61]	Luminal epithelial cells	SM16:0 (-/16:0), PC34:0 (16:0/18:0), PC34:1 (16:0/18:1), PC36:1 (18:0/18:1), PC34:2 (16:0/18:2), PC36:2 (18:0/18:2), PC36:4 (16:0/20:4), PC38:4 (18:0/20:4), PI34:0 (16:0/18:0), PI40:6 (18:0/22:6)	↑
		PC32:0 (16:0/16:0), PC40:6 (18:0/22:6), PE34:1 (16:0/18:1)	↓
Uterus (mouse), day 5 [61]	Uterine stroma cells	SM16:0 (-/16:0), PC34:0 (16:0/18:0), PC34:1 (16:0/18:1), PC36:1 (18:0/18:1), PC34:2 (16:0/18:2), PC36:2 (18:0/18:2), PC36:4 (16:0/20:4), PC38:4 (18:0/20:4), PC40:6 (18:0/22:6), PI34:2 (16:0/18:2), PI36:2 (18:0/18:2), PI40:6 (18:0/22:6)	↑
		PC32:0 (16:0/16:0)	↓
Uterus (mouse), day 6 [61]	PDZ	SM16:0 (-/16:0), PC34:1 (16:0/18:1), PC36:1 (18:0/18:1), PC34:2 (16:0/18:2), PC36:2 (18:0/18:2), PC36:4 (16:0/20:4), PC38:4 (18:0/20:4), PC40:6 (18:0/22:6),	↑
		PC32:0 (16:0/16:0)	↓
	M pole	PC34:1 (16:0/18:1), PE34:1 (16:0/18:1), PI38:5 (18:1/20:4)	↑
	AM pole	PC36:2 (18:0/18:2), PI34:2 (16:0/18:2)	↑
Uterus (mouse), day 6 [62]	PDZ	LPC18:0	↓
	PDZ	FA18:2	↑
	AM pole	PC36:2	↑
Uterus (mouse), day 7 [61]	PDZ	PC32:0(16:0/16:0)	↓
	M pole	PC34:0 (16:0/18:0), PC34:1 (16:0/18:1), PE34:1 (16:0/18:1), PI38:5 (18:1/20:4)	↑
	AM pole	PC34:2 (16:0/18:2), PC36:2 (18:0/18:2), PC36:4 (16:0/20:4), PC38:4 (18:0/20:4), PC40:6 (18:0/22:6), PI34:2 (16:0/18:2), PI36:2 (18:0/18:2), PS36:2 (18:0/18:2), PI40:6 (18:0/22:6)	↑
Uterus (mouse), day 8 [61]	M pole	PC34:1(16:0/18:1), PE34:1 (16:0/18:1), PG 34:1 (16:0/18:1), PS36:1 (18:0/18:1)	↑
	AM pole	PC34:2 (16:0/18:2), PC36:2 (18:0/18:2), PC36:4 (16:0/20:4), PC38:4 (18:0/20:4), PC40:6 (18:0/22:6), PI34:2 (16:0/18:2), PI36:2 (18:0/18:2), PS36:2 (18:0/18:2), PI40:6 (18:0/22:6)	↑
Uterus (mouse), day 8 [63]	M pole	PC34:1, PC36:1 (18:0/18:1), PC 36:3 (18:1/18:2)	↑
	AM pole	PC36:2 (18:0/18:2), PC38:4, PC40:6, DG34:2, DG36:4, DG36:3, DG36:2, DG38:4	↑

↑: increased; ↓: decreased; AM pole: antemesometrial pole; Cer: ceramide; DG: diacylglycerol; FA: fatty acid; LPC: lysophosphatidylcholine; M pole: mesometrial pole; PA: phosphatidic acid; PC: phosphatidylcholine; PDZ: primary decidual zone; PE: phosphatidylethanolamine; PG: phosphatidylglycerol; PI: phosphatidylinositol; PS: phosphatidylserine; SM: sphingomyelin.

## Data Availability

Not applicable.

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
