# Peer review of "Lipid Changes in the Peri-Implantation Period with Mass Spectrometry Imaging: A Systematic Review"

_life, 2023, doi:10.3390/life13010169_

Round 1
Reviewer 1 Report
Investigation of lipid changes in the peri-implantation period is useful to discover related biomarkers or reveal lipid metabolism with peri-implantation. Mass spectrometry, of cause, is the mainstream tool for lipidomics investigation for different samples and biological systems. However, that refers to LC-ESI-MS or direct infusing MS, but not MS imaging methods. 1) MS imaging methods usually lack good quantitative performance that is necessary for biomarker discovery; 2) there are not enough literature for about MS imaging of sample biological system for the authors to draw solid conclusions. This can be worse when the reproducibility of the MSI results are poor.
Author Response
Thank you for your work and kind comments. Yes, we totally agree, only good performed MSI can give us relevant information about possible biomarkers and a big limitation of this technique is that only gives us an m/z, so without fragmentation we cannot identify these molecules. In the discussion part we focus more on the limitation of MSI technique.
Reviewer 2 Report
Specific comments for the authors
* Abstract, Line 24, the sentence is not understood, especially what is meant by "among others". It can be written more clearly.
* Introduction, line 44, the sentence "The name 'Imaging' was first used by Caprioli and her research group in 1997" can be removed from the article.
* Introduction, lines 88-89. Why are lipid metabolism and signals important in embryo development or ovarian stimulation? This part should be explained briefly.
*Discussion, lines 265-266 (“The main advantage of MSI analysis compared to MS methods is that lipids can be analyzed in space without extensive preparation and without distorting the section.”), can be moved to the end of sentence line 259.
* Conclusions, I think it would be better if the authors briefly explained what useful contribution this lipid exchange mapping with MSI would make to the outcome for the peri-implantation period.

Author Response
We would like to thank your work and great suggestions that helped us to make our manuscript better. We especially thank you for the good overall rating. Below we answer the questions/comments.
* Abstract, Line 24, the sentence is not understood, especially what is meant by "among others". It can be written more clearly.
Answer: Thank you for this comment, we reworded this sentence.
* Introduction, line 44, the sentence "The name 'Imaging' was first used by Caprioli and her research group in 1997" can be removed from the article.
Answer: The sentence was deleted.
* Introduction, lines 88-89. Why are lipid metabolism and signals important in embryo development or ovarian stimulation? This part should be explained briefly.
Answer: The importance of lipid metabolism and signal in embryo implantation is now explained more detailed (rows 94-108).
*Discussion, lines 265-266 (“The main advantage of MSI analysis compared to MS methods is that lipids can be analyzed in space without extensive preparation and without distorting the section.”), can be moved to the end of sentence line 259.
Answer: The sentence was moved to the end of the previous section.
* Conclusions, I think it would be better if the authors briefly explained what useful contribution this lipid exchange mapping with MSI would make to the outcome for the peri-implantation period.
Answer: Thank you for this suggestion. Based on this comment we wrote a section (row 497-505)) about possible usefulness and implementation of MSI in the peri-implantation period.
Reviewer 3 Report
Lipids, like proteins and carbohydrates, belong to a category of biological molecules that perform multiple important functions in the biological systems. Structurally, lipids form the cellular barriers to segregate the internal and external cellular environment, and compartmentalize the intracellular space for specific functions. Metabolically, lipids are utilized as an important fuel source by various types of cells. Metabolism of lipids shuttles its energy storage to the core of biochemical reactions and produce energy currency that propels various cellular functions. Conversely, lipids also provide a metabolic flexibility by storing excessive energy in the form of hydrocarbon chain when energy intake exceeds demands, and serve as the regulatory factor of cellular energetics. The membrane lipids also form the heterogeneous patches of molecular scaffolds that promote the interactions between membrane proteins in signal transduction. Lipids and their metabolites are also used as the messengers in various signal transduction cascades. The importance of lipids in cancer biology and pathophysiology continue to capture the attention of the biomedical research communities and becomes an actively pursued subject under the “omics”-oriented studies.
Mass spectrometry imaging (MSI) of lipids within tissues has significant potential for both biomolecular discovery and histopathological applications. Conventional MSI technologies are, however, challenged by the prevalence of phospholipid regioisomers that differ only in the location(s) of carbon−carbon double bonds and/or the relative position of fatty acyl attachment to the glycerol backbone (i.e., sn position). The inability to resolve isomeric lipids in imaging experiments masks the underlying complexity, resulting in a critical loss of metabolic information. A viable alternative is ozone-induced dissociation (OzID) implemented on a quadrupole time-of-flight (Q-TOF) mass spectrometer capable of matrix-assisted laser desorption/ionization (MALDI). Exploiting the ion mobility region in the Q-TOF, high number densities of ozone were accessed, leading to ∼1000-fold enhancement in the abundance of OzID product ions compared to earlier MALDI-OzID implementations. Translation of this uplift into imaging resulted in a 50-fold improvement in acquisition rate, facilitating large-area mapping with resolution of phospholipid isomers. Mapping isomer distributions across tissue sections revealed distinct distributions of lipid isomer populations with region-specific associations of isomers differing in double bond and sn positions. Moreover, product ions arising from sequential ozone- and collision-induced dissociation enabled double bond assignments in unsaturated fatty acyl chains esterified at the noncanonical sn-1 position. A subjective data clustering analysis of MALDI-MSI result permitted the grouping of lipid mass spectrometric features into spatially salient segments in the tissue section, and assisted the discovery of several hidden yet significant mass spectrometric features. However also MALDI MSI technology presents limitations in applications due to pixel size in MALDI-MSI e.g. the area of desorption/ionisation on the tissue surface, that, in turn, is dependent on the laser spot size. Several groups have reported optical modifications to commercial MALDI-MSI ion sources that have reduced the laser spot size and, thus, achievable pixel sizes, down to < 10 μm and at best ~ 1 μm.
The paper covers an interesting topic but has several limitations. The authors point out the low number of articles included and analyzed. This reduces the scientific contribution of the results described and makes them obviously conditional in terms of statistical significance. Another limitation is that MSI technology, while representing an advance over MS, is nevertheless not the technique of choice for the study of lipidomics, which is much better approached with MALDI-MSI and derived techniques. However, the procedure followed for classification was well executed. I believe that this work can be accepted, but after a critical discussion of the limitations of the MSI technique compared to recent better techniques.
Author Response
We would like to thank the reviewer for the critical review process and comments, that helped us to improve our manuscript. Now in the discussion part (row 325-347) we included a whole section about the limitation of MSI techniques and compared the traditional techniques to some recent ones. We hope that now this section gives a better overview of this topic.
Reviewer 4 Report
Lipid changes in the peri-implantation period with mass spectrometry imaging: a systematic review
The authors have clearly done an extensive literature review to find relevant (mammalian) mass spectrometry imaging-based embryo implantation studies that can be used to potentially glean information on the human embryo implantation process. This review article will be helpful to researchers hoping learn more about this important topic. This article is acceptable for publication in Life after the following minor edits:
1) Page 2, lines 58-60 – While there can be difficulty identifying lipids from imaging studies, they are not difficult to detect. Lipids can even be detected without the use of solvent for matrix applications. One reason that researchers have shied away from lipid studies is the inability to perform knockout studies without lethality as many lipids are highly ubiquitous.
2) Page 11, lines 398-400 – While a couple of the authors are the same in the mouse studies, the mass spectrometry work was done in two different labs, the MALDI in one and the nano-DESI in another. The statement that all the studies were done in the same lab is a bit of a simplification.
Author Response
We would like to thank the reviewer for the work and valuable insights to help us improve the article. We especially appreciate the good overall rating. We have made several changes to the original manuscript based on your suggestions. We respond to each of the questions/suggestions below.
Suggestion 1: Page 2, lines 58-60 – While there can be difficulty identifying lipids from imaging studies, they are not difficult to detect. Lipids can even be detected without the use of solvent for matrix applications. One reason that researchers have shied away from lipid studies is the inability to perform knockout studies without lethality as many lipids are highly ubiquitous.
Answer: Thank you for your comment. We deleted the word ‘detection’ from the text.
Suggestion 2: Page 11, lines 398-400 – While a couple of the authors are the same in the mouse studies, the mass spectrometry work was done in two different labs, the MALDI in one and the nano-DESI in another. The statement that all the studies were done in the same lab is a bit of a simplification.
Answer: Thank you for bringing this simplification to our attention, and we have corrected the text and added a sentence to this section.
Round 2
Reviewer 1 Report
The authors tried to improve the quality of the manuscript. However, the conclusion could not be solid because MS imaging methods usually lack good quantitative performance that is necessary for biomarker discovery, and there are not enough literature for about MS imaging of sample biological system.
Author Response
Thank you for your comments. We have rephrased many sections in the introduction and tried to show the strenght and limitations of MSI in lipidomic studies. Our main aim with this systematic review was not the quantitative analysis of lipids, but detecting molecules in local section and see whether they could be used as a biomarker.
Reviewer 3 Report
I appreciated the authors' efforts to answer the referees' questions. The article, although it does not meet the criteria for novelty, may in any way be a valuable contribution in lipidomics studies. It can be accepted in the present version.
Author Response
Thank you again for your work and comments that helped us to improve our manuscript.